# Using Artificial Neural Network Models (ANNs) to Identify Patients with Idiopathic Normal Pressure Hydrocephalus (INPH) and Alzheimer Dementia (AD): Clinical Psychological Features and Differential Diagnosis

**DOI:** 10.3390/medicina61081332

**Published:** 2025-07-23

**Authors:** Lara Gitto, Carmela Mento, Giulia Massini, Paolo Massimo Buscema, Giovanni Raffa, Antonio Francesco Germanò, Maria Catena Ausilia Quattropani

**Affiliations:** 1Department of Economics, University of Messina, 98125 Messina, Italy; lara.gitto@unime.it; 2Department of Biomedical and Dental Sciences and Morphofunctional Imaging, University of Messina, 98125 Messina, Italy; giovanni.raffa@unime.it (G.R.); antonino.germano@unime.it (A.F.G.); 3Semeion Research Center of Sciences of Communication, 00128 Rome, Italy; g.massini@semeion.it (G.M.); m.buscema@semeion.it (P.M.B.); 4Department of Mathematical and Statistical Sciences, University of Colorado, 1201 Larimer Street, Denver, CO 80204, USA; 5Department of Educational Sciences, University of Catania, 95124 Catania, Italy; maria.quattropani@unict.it

**Keywords:** idiopathic normal pressure hydrocephalus (INPH), Alzheimer’s dementia (AD), artificial neural network (ANN) models, Rorschach inkblot test

## Abstract

*Background and Objectives*: Patients with idiopathic normal pressure hydrocephalus (INPH) present similar symptoms as other diseases, such as dementia (AD). However, while dementia is not reversible, INPH dementia can be treated through neurosurgery. This study aims to assess the Rorschach method as a valid tool to identify INPH patients. *Materials and Methods*: The perception characteristics of a small sample of patients (*n* = 19) were observed through the Rorschach Inblok test. Artificial neural network (ANN) models allowed us to analyze the correlations between patients’ cognitive functions and perception characteristics. *Results*: The results obtained revealed significant insights about the independent traits in patients’ patterns of response with INPH and AD. In performing the test, patients with INPH and AD concentrated more on the cards displayed and what they perceived, while other patients concentrated on reactions related to the image proposed. *Conclusions*: The Rorschach test is a valid predictor tool to identify INPH patients who could successfully be treated with neurosurgery. Hence, this methodology has potential in differential diagnosis applied to a clinical context.

## 1. Introduction

Idiopathic normal pressure hydrocephalus (INPH) is a brain disorder characterized by the enlargement of the brain’s ventricular system and cerebrospinal fluid (CSF) pressure within the normal range. INPH mainly affects men over 65 years of age; it is a progressive, chronic disorder causing cognitive impairment without a specific identifiable cause [1]. The idiopathic form must be distinguished from other chronic forms of hydrocephalus conditions, which can derive from brain surgery, meningitis, or head injury in 50% of cases [2].

INPH is characterized by ventricular enlargement without intracranial hypertension, which causes clinical gait disturbances, urinary incontinence, and dementia (known as the “Adams triad”) and accounts for 2–10% of all forms of dementia and 40% of adulthood hydrocephalus.

The classic triad is similar in appearance to other neurological conditions, such as Alzheimer’s Dementia (AD) [1]. However, while dementia is not reversible, INPH dementia can be treated through neurosurgical procedures by installing a shunt to drain excess CSF into another part of the body. This treatment may alleviate symptoms and help restore normal cerebrospinal fluid dynamics [3].

In fact, after treatment, clinical improvements have been reported in 30–96% of patients [4]. It was found that 93% of patients experience gait improvements, and more than 50% undergo significant cognitive improvements [5,6].

Despite a large amount of data available, some aspects concerning the differential diagnosis of various forms of dementia and the accuracy of patient selection for ventriculoperitoneal shunts are still controversial [7].

INPH is a significant challenge for the medical community [8]: it is extremely difficult to differentiate between INPH and AD based on neurological deficits and clinical evidence only, such as measuring ventricular size through conventional medical imaging techniques [9]. The distinction is clinically important because INPH is one of the few treatable causes of dementia [10].

Hence, the main costs of INPH derive from its lack of diagnosis. Recently, this issue was analyzed in a conference held at the Italian Ministry of Health (https://www.sanitainformazione.it/idrocefalo-normoteso-attenzione-a-non-confonderlo-con-alzheimer-o-parkinson/, accessed on 7 May 2025). The findings of this conference confirmed the urgent attention this issue requires. Moreover, since there is no formal definition of INPH, discrepancies are encountered when estimating its real incidence, which is thought to range from 2 to 20 per million/people per year. The difficulty of distinguishing INPH from other neurodegenerative disorders is the most likely reason why about 80% of cases remain unrecognized and untreated [11]. Although data report a total of 5/100,000 new cases diagnosed every year, the actual number of people with INPH is predicted to be higher. A percentage ranging from 9 to 14% of the elderly living at home present with INPH symptoms, and a further increase in the number of people suffering from this disease can be assumed considering population aging.

Accurate diagnosis is necessary because conventional hydrocephalus treatments have no benefit in non-hydrocephalus patients: if non-hydrocephalic patients are misdiagnosed as hydrocephalic patients, treatment is not only ineffective but associated with significant morbidity and increased costs [12].

In light of these considerations, since the overall prevalence of dementia rises progressively as the population ages, even if INPH is responsible for only a small proportion of senile dementia, successful treatment could help many patients [13]. They could be treated in a short time and save on resources; costs due to incorrect therapies or inappropriate diagnostic procedures would be lower, and patient’s quality of life would be higher.

Regarding the type of treatment and assistance used, the spontaneous course of INPH requires most patients to rely on nursing care [14]. For those patients undergoing surgery, continuous and regular checks are required by a neurosurgeon to ensure that the shunt is working properly [15]; continuous follow-up and assistance can be complemented with physiotherapy treatments, socialization, and psychological support.

A comprehensive estimation of INPH costs has not been performed as of yet. In 2000, more than two decades ago, the cost of treating INPH in the US exceeded USD 1 billion; there were 27,870 patients with treated INPH, while more than 8000 were newly cases diagnosed. However, updated statistics are necessary [16].

Together with considerations of costs, another field of analysis is the implementation of new diagnostic tools. To our knowledge, no clinical or neuroradiological techniques have been validated to clearly identify dementia from INPH, both of which share anatomical and clinical similarities [17]. Magnetic resonance imaging (MRI) depicts ventricle size accurately; however, findings on brain images are not sufficient to establish a diagnosis alone because they provide minimal, if any, evidence of brain damage despite marked deficits in motor skills and cognitive functioning [10,18,19].

In a study describing the advantages and drawbacks of MRI parameters, the importance of finding specific imaging biomarkers likely to distinguish between the two conditions has been emphasized [20]. If a screening tool can identify possible cases, then further workup should be performed to confirm the diagnosis and determine the need for shunting [21,22,23].

The Rorschach test is usually involved in psychological diagnosis; clinical, personality assessments; and the selection context for the detection of a global personality profile in patterns of response [24,25,26,27,28]. The response process of the inkblot is centered on the identification of mental function, cognitive pattern, and characteristic traits of personality [29,30]. The answers to the questions posed through the perceptual task achieve patterns of cognition alongside visual perception, language processing, or the inhibition of phases of the response process [31,32,33,34]. In this study, the Rorschach test is not only a substitute for clinical diagnosis but an exploratory tool to identify potential response patterns that may help generate hypotheses and support the differential diagnosis between iNPH and AD.

Despite the pioneering nature of the present study, its results must be carefully considered since they allow common traits and patterns of response to be identified in patients with INPH and AD.

Hence, this study aims to assess the Rorschach method as a valid tool for identifying INPH patients. The information collected by administering the test is analyzed through the implementation of three different methods: Population, Self-Organizing Maps (SOMs), and artificial neural networks (ANNs). The characteristics of INPH vs. AD patients are outlined and commented on. Some comments on the advantages of adopting the Rorschach test in complex diagnoses will conclude the study. The innovative aspect of this work lies in the proposal of an alternative methodology to support the identification of patients presenting with cognitive decline. Importantly, the findings are intended to generate hypotheses and do not serve a confirmatory diagnostic purpose.

## 2. Materials and Methods

A small sample of patients (*n* = 19) was observed at the Polyclinic of Messina, Southern Italy, in the year 2015. Eleven patients suffered from INPH, while the remaining 8 presented with AD dementia. The observed patients (or their caregivers) provided written informed consent, and the study was conducted in accordance with the Declaration of Helsinki and approved by the Ethics Committee at the University Hospital of Messina (Prot. 28/219, 26 November 2014).

Two psychological tests were carried out.

The patient’s mental status was measured through the Mini-Mental State Examination (MMSE) [35], an 11-item questionnaire testing areas of cognitive function (orientation, registration, attention, calculation, recall, and language). A low score is indicative of cognitive impairment.

Perception characteristics and personality traits were observed through the Personality Rorschach Inkblot test [26,28,36,37,38,39]. The test comprises 10 symmetrical inkblots: 5 monochrome, 2 two-tone, and 3 colored inkblots. The images are brought to the attention of the subject, one by one; there is no time limit imposed, and each response comprises an interpretation of the characteristics of each image, i.e., its form, content, and the determinants, codified as form, color, shading, and movement [40,41,42,43]. The coding of the responses is based on the time needed to provide or refuse an answer and provide any additional comments [28,30].

The information provided on the general aspects of the inkblot (shape, color, etc.) and the location (such as its details) are often considered more important than the content itself; considering the originality of the response, this is considered positive or negative (+ or −) in relation to good and poor form [37]. Considering the theme of the image, the content (whether it is human, natural, animal, abstract, etc.) is classified into categories according to the frequency of the interpretation [26,28,36]; the emotional life of the subject is centered on the colors and its shading [41,43].

The literature provides an example of the clinical application of the inkblot task in a sample of elderly subjects or suspected dementia patients [30,31] for the differential diagnosis of brain impairment [44,45]. Organic signs were also observed in brain subcortical and cortical involvement and in cognitive deficiency in the performance that occurred in organic lesions [33,44,46,47]. The Inkblot task can detect perceptive and visuo-spatial functions from a neuropsychological perspective in a clinical assessment [48,49,50].

The information collected from the administration of the Rorschach Inkblot Test was analyzed using computational methods applying different algorithms.

These methods are as follows and in the Appendix A:-**Populations**: This is a Multi-Dimensional Scaling algorithm that projects the observed individuals on a bi-dimensional plane with coordinates X and Y [51];-**Self-Organizing Map (SOM)**: This is an example of an unsupervised neural network, clustering the observed individuals and focusing on input variables on a bi-dimensional matrix [52];-**Auto Contractive Map (AutoCM)**: This is an unsupervised neural network that detects existing relationships between variables based on the values that can be attributed to everyone; the results can be represented through a graph (the *Maximally Regular Graph,* MRG) [53,54].

## 3. Results

The descriptive statistics of the sample examined can be observed in Table 1.

The mean age of the observed patients was 76 years old (min 59 years; max 88 years old). The sample included 11 males and 8 women; all male patients, apart from one, suffered from INPH. Twelve patients were older than 75 years, and 89% were married. The average number of answers was 10.32 for the ten tables (min 4; max 18).

Despite the small number of subjects in the sample, the methods employed for the analysis allowed the two groups of patients to be clustered.

### 3.1. Results with Populations

In Table 2, the input data employed in the calculations carried out with *Populations* are shown. On each line, 19 subjects (also called “cases” or “records”) are reported; on each column, 45 variables are employed.

The *Populations* method organized the 19 subjects on a two-dimensional plane. Each subject is, therefore, represented by the point with coordinates x and y. The distance between the points is the difference between the values of the 45 variables in the records.

The distribution obtained shows that, except for one subject (which we will refer to as “dementia 1”), the system distinguished between the two populations of patients (Figure 1).

Hence, according to this algorithm, the variables included in the analysis are sufficient for the identification of the two types of subjects.

### 3.2. Results with SOM Algorithm

An SOM was set up with an output matrix measuring 5 × 5; therefore, it was potentially possible to obtain 25 classes.

Given the reduced number of records (19) out of 25 “virtual” classes, SOM located each subject in a class. Only in one case did two subjects share the same class (in the output, this was class 5.2; the subjects were INPH9 and INPH7).

In Figure 2, it can be noted how the two populations were distributed in different areas of the matrix.

The results confirm the findings obtained using the *Populations* method; therefore, the 45 variables are good indicators to distinguish between the two pathologies: INPH vs. AD. More specifically, when processing the data, some variables, among which there were, for example, the determining variables “Global”, “Detail”, “Form”, “Good form”, etc., did not show a specific distribution. Instead, other variables, such as “Form over diffuse shading”, “Original”, “Original characterized by good form”, and “Inadequacy”, showed a high value in some classes only.

### 3.3. Results with AutoCM

Lastly, the AutoCM model is helpful in understanding the relationship between each variable and the two populations of patients.

The web input was set up using the same 45 variables included in the two previous experiments by adding two variables that discriminated between subjects according to their pathology (see the last column of Table 3); in the graphical representation, the characteristics of the two pathologies are highlighted.

The AutoCM learned the records made by the 19 subjects. Every layer of the web (input, hidden, and output) was composed of 47 units. When the processing was finished, the most relevant connections among the 47 variables were highlighted by the MRG.

Figure 3 and Figure 4 show the graphs obtained with the ANNs, respectively, both with and without the values showing the relationship among variables. Some variables connected to Rorschach’s interpretation are more closely linked to the diagnosis of INPH and AD.

We can summarize the evidence obtained by these graphical representations.

For the INPH group, the most relevant variables were animal contents (A); form response (F); details (D); popular response; and adequacy of index of reality (IR).

In AD, the most relevant variables were perseveration phenomena, inadequacy, and particular phenomena, such as card rejection and linguistic errors.

That means that INPH (and, consequently, patients’ behavior) is strictly linked (connection value = 0.99) to animal forms, indicating higher emotional inhibition to common responses (popular) and well-formed responses (connection value between popular responses and forms = 0.99). The number of responses, strictly linked to shape, was also connected to the perception of details in the interpretation, well-formed responses, and the reality index.

The data shows how the structure highlighted in this typology of patients has better adaptation to reality with perception levels closer to the average population in comparison to psychopathological phenomena observed in AD cases (i.e., perseveration, card rejection, and linguistic errors).

Instead, patients with AD are more closely linked to phenomena highlighting the behavior of subjects in relation to the stimulus represented on the board (i.e., particular phenomena, for example, waste (0.99); subjects’ reactions are uncertainty (0.99) and preserved (0.98). The latter is connected to inadequacy in the response.

## 4. Discussion

This study highlights the implications of using the Inkblot task in clinical practice and neuropsychological assessments for the differential diagnosis of patients suffering from organic and neurodegenerative diseases, such as Idiopathic Normal Pressure Hydrocephalus (INPH) and Alzheimer’s disease (AD).

As specified in the Introduction, INPH is a syndrome characterized by gait impairment, cognitive decline, and urinary incontinence and is associated with ventriculomegaly in the absence of high cerebrospinal fluid (CSF) pressure [55]. It is different from AD, which is a chronic neurodegenerative disease that usually worsens slowly over time. 

The Rorschach method can be considered a neuropsychological method to detect alterations in psychic functions [28,37,46,48]. In other contributions, alterations in psychic functions have been studied in relation to memory deficits, poor emotional and impulse control, linguistic errors, and processes of response [31,32,34,44]. In cognitive impairment, elderly patients are unable to synthesize perceptual details or the organization of complex forms, which is centered on the deficit of visual perception, recognition, a lack of awareness, and signs of organicity [32,46]. Unlike patients with AD, those with INPH present an interpretative awareness and do not make linguistic errors. Other authors highlight the potential of this test in neuropsychological assessments related to response processes and psychic functions as well as perceptual processing, attention, memory, and executive functioning [32,45,48,49,50].

The first analysis technique employed showed how the two subsamples of patients are distinct. Then, we processed data with the SOM. This technique, in line with the literature, was a good indicator to distinguish between the two groups of patients.

In addition, the AutoCM model was useful in understanding the relationship between variables (such as animal contents (A); form response (F); details (D); popular response; and index of reality (IR)) that are strictly connected to INPH diagnosis. Variables connected to AD diagnosis can be identified in perseveration phenomena, inadequacy, particular phenomena, card rejection, and linguistic errors. As far as INPH is concerned, higher emotional inhibition can be found in animal responses along with a higher adaptation rate to reality, which differs from AD patients. These patients have a good adaptation to reality and retain perception levels closer to the average population compared to the psychopathological signs that are observed in patients with AD, i.e., rejection, perseveration processes, and linguistic mistakes. The neuropsychological aspects of patients with cognitive deficits include an inability to grasp the theme regarding the difficulty of reenacting memory and a low control of pulses, usually identified as C, CF, or extra-tensive resonance, codified in the form of subjective Erlebnistypus [26,32,43,45,46].

The results obtained through the implementation of ANNs demonstrate the potential of the Rorschach test to identify patients presenting with INPH, even within a small sample. During the assessment, patients concentrated more on what the table displayed and their perceptions, while AD patients concentrated on personal reactions (such as particular phenomena) in relation to the image used.

The group INPH did not show clear signs of organic dysfunction, as is found specifically in people with AD. The cognitive patterns connected to AD, as stated in the literature, include signs of perplexity in the interpretation process, the perseveration of the content, visuospatial deficit, and language mistakes in the response process [32,44,50].

This evidence shows that the Rorschach method is a cognitive task involving the global brain in the response process and is suitable for detecting perceptual deficits, including visual perception patterns, object recognition, and language production [30,34].

## 5. Conclusions

A neuropsychological approach to clinical assessments represents a milestone for empirical research in differential diagnosis from a clinical viewpoint.

The neuropsychological approach based on the signs of the Inkblot task has been shown to distinguish between two different diagnoses. Overall, the test detected mental alterations linked to each disease, identifying a pattern of cognitive functions (such as signs of memory impairment, recognition, emotions and control impulses, visual attention, and executive functioning, and language processes).Since it can be used on all individuals without the limitations of education level, this test is a valid predictor task in broader neuropsychological evaluations and is able to identify the cognitive patterns of INPH patients.

The three distinct methods used led to the same conclusions. Even if carried out on a small sample, the present study suggests original and innovative results in the assessment and clinical research of reversible dementia. If correctly and timely diagnosed, INPH can be corrected through neurosurgical treatment, achieving an improvement in patients’ wellbeing and quality of life, together with a reduction in the cost of illness.

## Figures and Tables

**Figure 1 medicina-61-01332-f001:**
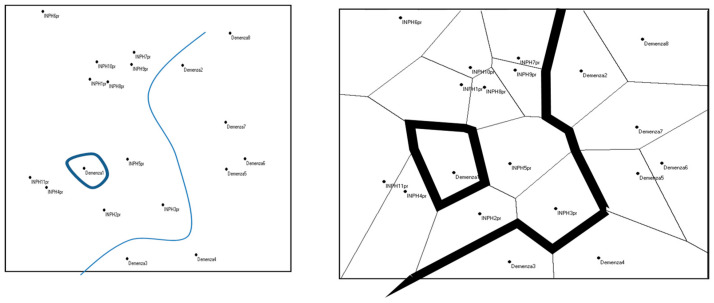
Results with the *Population* algorithm.

**Figure 2 medicina-61-01332-f002:**
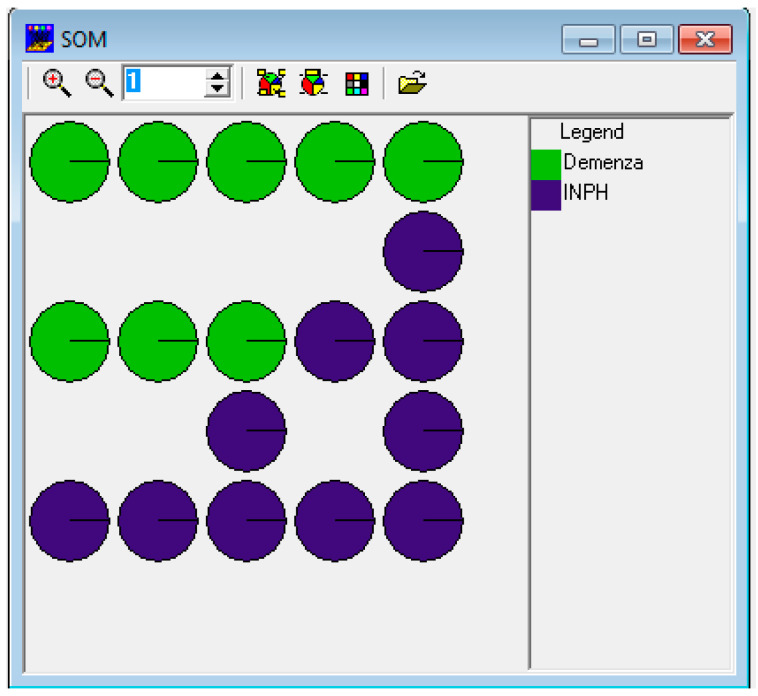
The results with SOM.

**Figure 3 medicina-61-01332-f003:**
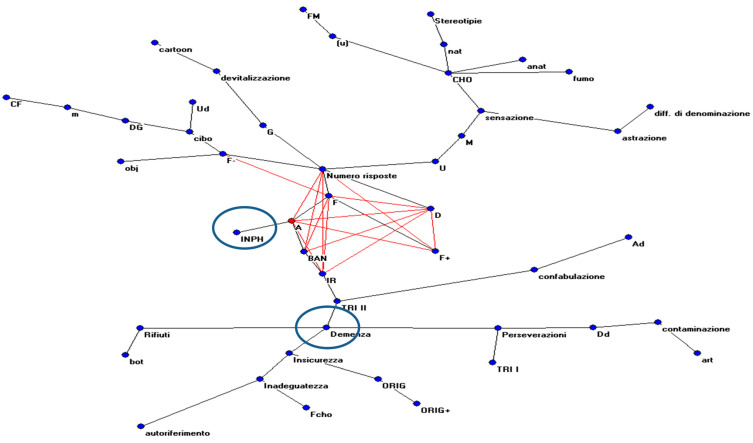
The MRG obtained by analyzing the connections between the hidden and output units of the ANNs.

**Figure 4 medicina-61-01332-f004:**
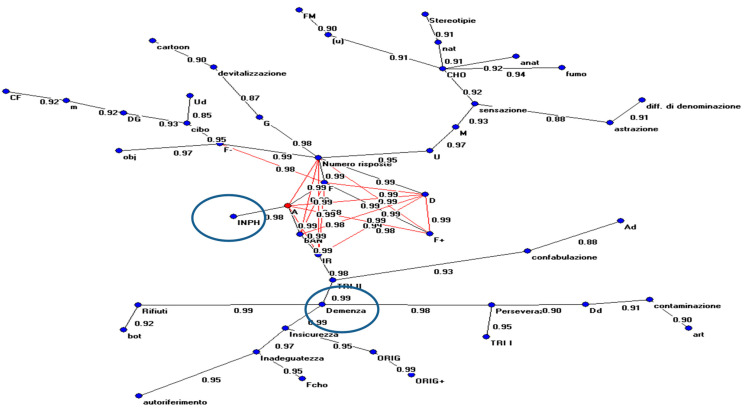
The MRG obtained by analyzing the connections between the hidden and output units of the ANNs (with indications for each value).

**Table 1 medicina-61-01332-t001:** Descriptive statistics.

Variable	Mean	Std. Dev.	Min	Max
Age	76.11	7.32	59	88
Age > 75	0.58	0.51	0	1
Gender (1 = female; 0 = male)	0.58	0.51	0	1
Compulsory education	0.63	0.50	0	1
Married (1 = yes; 0 = no)	0.89	0.32	0	1
Number of answers	10.32	3.83	4	18
MMSE total score	17.89	5.15	8	27
Localization:				
Global (G)	5.42	2.81	1	13
Details (D)	4.58	3.06	0	10
Little details (Dd)	0.11	0.32	0	1
Details → global (DG)	0.21	0.71	0	1
Determinants:				
F (shape)	8.53	4.14	2	16
F+ (positive shape)	5.63	3.20	1	12
F− (negative shape)	2.89	2.21	0	8
M (Human kinesthetic activity)	0.74	0.73	0	3
FM (Animal kinesthetic activity)	0.74	1.24	0	4
m (inanimate movement)	0.05	0.23	0	1
CF (color and shame)	0.05	0.23	0	1
FCho (diffuse shading)	0.11	0.32	0	1
Cho (pure shading)	0.05	0.23	0	1
Contents:				
A (animals)	5.84	2.59	0	10
Ad (animal details)	0.47	1.02	0	4
H (human)	1.26	1.48	0	5
(h) (fantastic human)	0.11	0.32	0	1
Hd human details	0.37	0.83	0	3
Obj object	0.95	1.31	0	4
Cibo food	0.16	0.37	0	1
Nat nature	0.11	0.32	0	1
Botanic	0.58	0.96	0	3
Anatomical	0.32	0.75	0	3
Smoke	0.05	0.23	0	1
Cartoon	0.11	0.32	0	1
Art	0.11	0.32	0	1
*Popular*	2.21	1.17	0	4
*Original*	0.11	0.32	0	1
TRI I (Erleibniss typus I)	1.74	0.99	1	3
TRI II (Erleibniss typus II)	2.16	0.96	1	3
IR (reality index)	3.58	1.80	0	6
Particular phenomena				
Perseveration	0.37	0.50	0	1
Inadequacy	0.21	0.42	0	1
Insecurity	0.26	0.45	0	1
Stereotyped	0.05	0.23	0	1
Refuse	0.63	0.50	0	1
Confabulation	0.21	0.42	0	1
Abstraction	0.05	0.23	0	1
Contamination	0.05	0.23	0	1
Difficulty of naming	0.05	0.23	0	1
Self-reference	0.11	0.32	0	1
Devitalization	0.11	0.32	0	1
Sensation	0.11	0.32	0	1

**Table 2 medicina-61-01332-t002:** The data set used in machine learning. There are 19 subjects represented by the lines; 45 variables are represented in the columns.

Caso	Numero Risposte	G	D	Dd	DG	F	F+	F−	M	FM	m	CF	Fcho	CHO	BAN	ORIG	ORIG+	A	Ad	U	(u)	Ud	Obj	Cibo	Nat	Bot	Anat	Fumo	Cartoon	Art	IR	TRI I	TRI II	Perseverazioni	Inadeguatezza	Insicurezza	Stereotipie	Rifiuti	Confabulazione	Astrazione	Contaminazione	Diff. Denominazione	Autoriferimento	Devitalizzazione	Sensazione
Demenza 1	8	8	0	0	0	8	5	3	0	0	0	0	0	0	0	0	0	6	0	0	0	0	2	0	0	0	0	0	1	0	0	3	3	0	0	0	0	1	0	0	0	0	0	0	0
Demenza 2	7	2	5	0	0	5	1	4	1	0	0	0	1	0	2	0	0	5	0	1	0	0	0	0	0	1	0	0	0	0	4	1	3	0	1	0	0	1	1	0	0	0	0	0	0
Demenza 3	10	4	3	0	3	8	3	5	0	0	1	1	0	0	1	0	0	3	0	0	0	2	4	1	0	0	0	0	0	0	2	3	2	1	0	1	0	1	0	0	0	0	0	0	0
Demenza 4	15	7	8	0	0	14	12	2	1	0	0	0	0	0	4	1	1	9	0	3	0	0	2	0	1	0	0	0	0	0	5	1	3	1	0	1	1	1	0	0	0	0	0	0	0
Demenza 5	9	1	7	1	0	9	7	2	0	0	0	0	0	0	1	0	0	5	2	0	0	0	0	0	0	2	0	0	0	0	2	3	3	1	1	1	0	1	1	0	0	0	0	0	0
Demenza 6	13	3	9	0	1	11	8	3	1	0	0	0	1	0	3	1	1	7	1	2	0	0	0	1	0	2	0	0	0	0	4	1	2	0	1	1	0	1	0	0	0	0	1	0	0
Demenza 7	8	2	6	0	0	7	7	0	1	0	0	0	0	0	2	0	0	3	0	2	0	3	0	0	0	0	0	0	0	0	4	1	3	0	1	1	0	0	0	0	0	0	1	0	0
Demenza 8	7	5	2	0	0	4	2	2	3	0	0	0	0	0	1	0	0	0	0	5	0	0	0	0	0	2	0	0	0	0	2	1	3	1	0	0	0	1	1	1	0	1	0	0	1
INPH 1 pr	10	7	3	0	0	6	5	1	0	4	0	0	0	0	1	0	0	9	0	0	0	0	1	0	0	0	0	0	0	0	2	3	1	0	0	0	0	0	0	0	0	0	0	0	0
INPH 2 pr	13	5	8	0	0	13	7	6	0	0	0	0	0	0	4	0	0	7	4	0	0	0	1	0	0	0	1	0	0	0	6	3	3	0	0	0	0	0	1	0	0	0	0	0	0
INPH 3 pr	13	6	7	0	0	13	10	3	0	0	0	0	0	0	4	0	0	10	0	0	0	0	0	0	0	3	0	0	0	0	5	3	3	1	0	0	0	1	0	0	0	0	0	0	0
INPH 4 pr	16	13	3	0	0	15	7	8	1	0	0	0	0	0	4	0	0	6	1	2	0	1	3	1	0	0	1	0	1	0	6	1	3	0	0	0	0	0	0	0	0	0	0	1	0
INPH 5 pr	7	7	0	0	0	7	7	0	0	0	0	0	0	0	2	0	0	7	0	0	0	0	0	0	0	0	0	0	0	0	2	3	3	1	0	0	0	0	0	0	0	0	0	0	0
INPH 6 pr	12	7	5	0	0	7	3	4	1	3	0	0	0	1	3	0	0	6	0	0	1	1	0	0	1	0	3	1	0	0	4	1	1	0	0	0	0	0	0	0	0	0	0	0	1
INPH 7 pr	4	3	1	0	0	2	2	0	1	1	0	0	0	0	3	0	0	3	0	1	0	0	0	0	0	0	0	0	0	0	6	1	1	0	0	0	0	1	0	0	0	0	0	0	0
INPH 8 pr	11	6	5	0	0	8	4	4	1	1	0	0	0	0	1	0	0	6	1	1	0	0	2	0	0	0	0	0	0	1	2	1	1	0	0	0	0	1	0	0	0	0	0	0	0
INPH 9 pr	4	3	1	0	0	2	2	0	1	1	0	0	0	0	3	0	0	3	0	1	0	0	0	0	0	0	0	0	0	0	6	1	1	0	0	0	0	1	0	0	0	0	0	0	0
INPH 10 pr	11	7	4	0	0	7	4	3	1	3	0	0	0	0	2	0	0	7	0	2	0	0	0	0	0	1	1	0	0	0	4	1	1	0	0	0	0	1	0	0	0	0	0	1	0
INPH 11 pr	18	7	10	1	0	16	11	5	1	1	0	0	0	0	1	0	0	9	0	4	1	0	3	0	0	0	0	0	0	1	2	1	1	1	0	0	0	0	0	0	1	0	0	0	0

**Table 3 medicina-61-01332-t003:** The data set used as the input of the AutoCM network. Nineteen subjects are included in the rows; 47 variables are employed in the columns.

Caso	Numero Risposte	G	D	Dd	DG	F	F+	F−	M	FM	m	CF	Fcho	CHO	BAN	ORIG	ORIG+	A	Ad	U	(u)	Ud	Obj	Cibo	Nat	Bot	Anat	Fumo	Cartoon	Art	IR	TRI I	TRI II	Perseverazioni	Inadeguatezza	Insicurezza	Stereotipie	Rifiuti	Confabulazione	Astrazione	Contaminazione	Diff. Di Denominazione	Autoriferimento	Devitalizzazione	Sensazione
Demenza 1	8	8	0	0	0	8	5	3	0	0	0	0	0	0	0	0	0	6	0	0	0	0	2	0	0	0	0	0	1	0	0	3	3	0	0	0	0	1	0	0	0	0	0	0	0
Demenza 2	7	2	5	0	0	5	1	4	1	0	0	0	1	0	2	0	0	5	0	1	0	0	0	0	0	1	0	0	0	0	4	1	3	0	1	0	0	1	1	0	0	0	0	0	0
Demenza 3	10	4	3	0	3	8	3	5	0	0	1	1	0	0	1	0	0	3	0	0	0	2	4	1	0	0	0	0	0	0	2	3	2	1	0	1	0	1	0	0	0	0	0	0	0
Demenza 4	15	7	8	0	0	14	12	2	1	0	0	0	0	0	4	1	1	9	0	3	0	0	2	0	1	0	0	0	0	0	5	1	3	1	0	1	1	1	0	0	0	0	0	0	0
Demenza 5	9	1	7	1	0	9	7	2	0	0	0	0	0	0	1	0	0	5	2	0	0	0	0	0	0	2	0	0	0	0	2	3	3	1	1	1	0	1	1	0	0	0	0	0	0
Demenza 6	13	3	9	0	1	11	8	3	1	0	0	0	1	0	3	1	1	7	1	2	0	0	0	1	0	2	0	0	0	0	4	1	2	0	1	1	0	1	0	0	0	0	1	0	0
Demenza 7	8	2	6	0	0	7	7	0	1	0	0	0	0	0	2	0	0	3	0	2	0	3	0	0	0	0	0	0	0	0	4	1	3	0	1	1	0	0	0	0	0	0	1	0	0
Demenza 8	7	5	2	0	0	4	2	2	3	0	0	0	0	0	1	0	0	0	0	5	0	0	0	0	0	2	0	0	0	0	2	1	3	1	0	0	0	1	1	1	0	1	0	0	1
INPH 1 pr	10	7	3	0	0	6	5	1	0	4	0	0	0	0	1	0	0	9	0	0	0	0	1	0	0	0	0	0	0	0	2	3	1	0	0	0	0	0	0	0	0	0	0	0	0
INPH 2 pr	13	5	8	0	0	13	7	6	0	0	0	0	0	0	4	0	0	7	4	0	0	0	1	0	0	0	1	0	0	0	6	3	3	0	0	0	0	0	1	0	0	0	0	0	0
INPH 3 pr	13	6	7	0	0	13	10	3	0	0	0	0	0	0	4	0	0	10	0	0	0	0	0	0	0	3	0	0	0	0	5	3	3	1	0	0	0	1	0	0	0	0	0	0	0
INPH 4 pr	16	13	3	0	0	15	7	8	1	0	0	0	0	0	4	0	0	6	1	2	0	1	3	1	0	0	1	0	1	0	6	1	3	0	0	0	0	0	0	0	0	0	0	1	0
INPH 5 pr	7	7	0	0	0	7	7	0	0	0	0	0	0	0	2	0	0	7	0	0	0	0	0	0	0	0	0	0	0	0	2	3	3	1	0	0	0	0	0	0	0	0	0	0	0
INPH 6 pr	12	7	5	0	0	7	3	4	1	3	0	0	0	1	3	0	0	6	0	0	1	1	0	0	1	0	3	1	0	0	4	1	1	0	0	0	0	0	0	0	0	0	0	0	1
INPH 7 pr	4	3	1	0	0	2	2	0	1	1	0	0	0	0	3	0	0	3	0	1	0	0	0	0	0	0	0	0	0	0	6	1	1	0	0	0	0	1	0	0	0	0	0	0	
INPH 8 pr	11	6	5	0	0	8	4	4	1	1	0	0	0	0	1	0	0	6	1	1	0	0	2	0	0	0	0	0	0	1	2	1	1	0	0	0	0	1	0	0	0	0	0	0	0
INPH 9 pr	4	3	1	0	0	2	2	0	1	1	0	0	0	0	3	0	0	3	0	1	0	0	0	0	0	0	0	0	0	0	6	1	1	0	0	0	0	1	0	0	0	0	0	0	0
INPH 10 pr	11	7	4	0	0	7	4	3	1	3	0	0	0	0	2	0	0	7	0	2	0	0	0	0	0	1	1	0	0	0	4	1	1	0	0	0	0	1	0	0	0	0	0	1	0
INPH 11 pr	18	7	10	1	0	16	11	5	1	1	0	0	0	0	1	0	0	9	0	4	1	0	3	0	0	0	0	0	0	1	2	1	1	1	0	0	0	0	0	0	1	0	0	0	0

## Data Availability

The original contributions presented in this study are included in the article. Further inquiries can be directed to the corresponding author.

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
