# Peer review of "Using Artificial Neural Network Models (ANNs) to Identify Patients with Idiopathic Normal Pressure Hydrocephalus (INPH) and Alzheimer Dementia (AD): Clinical Psychological Features and Differential Diagnosis"

_medicina, 2025, doi:10.3390/medicina61081332_

Round 1
Reviewer 1 Report
Comments and Suggestions for Authors
The manuscript presents a novel intersection between artificial neural networks and clinical psychological profiling, focusing on the differential diagnosis between INPH and AD using Rorschach Inkblot tests. This is an intriguing and interdisciplinary approach. However, while the topic is innovative, there are significant methodological, analytical, and interpretive limitations that should be addressed before considering the paper for publication.
- clarify that findings are exploratory and hypothesis-generating, not confirmatory
- Rorschach test’s use as a diagnostic differentiator, especially for neurodegenerative diseases, is unorthodox.
- The visual separation of clusters in SOM or AutoCM is interpreted too liberally.
- How would this tool be integrated into diagnostic pathways? How does it compare to MRI, CSF tap test, or standard cognitive tests?
- No statistical significance testing or confidence intervals are presented.
- Consider simplifying the title, which is currently long and overly descriptive.
- The quality of Figures 3 and 4 (MRG graphs) is poor; axes are not labeled, and interpretation requires prior knowledge.
Author Response
Review 1
Dear Editor,
Thank you for the opportunity to revise our above-referenced paper. The following letter gives our responses to the Editor’s and Reviewers’ comments, as requested.
We wish to thank the Reviewers for reading our manuscript and reviewing it, which has helped us improve it. We next detail our responses to each reviewer’s concerns and comments.
Open Review
(x) I would not like to sign my review report
( ) I would like to sign my review report
Quality of English Language
( ) The English could be improved to more clearly express the research.
(x) The English is fine and does not require any improvement.
|
Yes |
Can be improved |
Must be improved |
Not applicable |
|
|
Does the introduction provide sufficient background and include all relevant references? |
( ) |
(x) |
( ) |
( ) |
|
Is the research design appropriate? |
( ) |
(x) |
( ) |
( ) |
|
Are the methods adequately described? |
( ) |
( ) |
(x) |
( ) |
|
Are the results clearly presented? |
( ) |
( ) |
(x) |
( ) |
|
Are the conclusions supported by the results? |
( ) |
(x) |
( ) |
( ) |
|
Are all figures and tables clear and well-presented? |
( ) |
( ) |
(x) |
( ) |
Comments and Suggestions for Authors
The manuscript presents a novel intersection between artificial neural networks and clinical psychological profiling, focusing on the differential diagnosis between INPH and AD using Rorschach Inkblot tests. This is an intriguing and interdisciplinary approach. However, while the topic is innovative, there are significant methodological, analytical, and interpretive limitations that should be addressed before considering the paper for publication.
clarify that findings are exploratory and hypothesis-generating, not confirmatory
The innovativeness of the work lies in the fact that an alternative methodology is being presented to identify whether patients showing symptoms of cognitive decline are affected by iNPH or AD. Obviously, findings may be helpful in generating hypotheses and are not confirmatory.
We are aware that Rorschach test’s use as a diagnostic differentiator, especially for neurodegenerative diseases, is unorthodox. Hence, it cannot substitute a clinical diagnosis.
#R to Rev1#: Thank you for your valuable comment. It was made clear in the manuscript that the results can be useful for generating hypotheses and are not confirmatory. Furthermore, we agree that the use of the Rorschach test as a diagnostic differentiator is unorthodox and cannot replace a clinical diagnosis.
The visual separation of clusters in SOM or AutoCM is interpreted too liberally.
Clearly, the observed sample is too small to draw conclusive observations. It is significant how a distribution among patients with differing characteristics is nonetheless detected.
How would this tool be integrated into diagnostic pathways? How does it compare to MRI, CSF tap test, or standard cognitive tests?
It is clear that such finding obtained through the Rorschach test can never replace an instrumental assessment. Rather, it can be considered as supportive of the diagnosis.
No statistical significance testing or confidence intervals are presented.
The objective of this first exploratory analysis is that of representing visually the two sub-samples identified through the ANNs. Hence, we concentrated on such evidence that we observe from the analysis.
Consider simplifying the title, which is currently long and overly descriptive.
The title has been shortened in “Artificial Neural Network Models (ANNs) in Identifying Idiopathic Normal Pressure Hydrocephalus (INPH) and Alzheimer Dementia (AD) Patients
The quality of Figures 3 and 4 (MRG graphs) is poor; axes are not labeled, and interpretation requires prior knowledge.
The graph is aimed at showing the connections that have been outined through the analysis carried out. There are not horizontal and vertical axes to consider: rather, the ANN identified can be placed in the space and what matter is the proximity of the variables considered and the level of significance of some of these connections,
Submission Date
09 June 2025
Date of this review
20 Jun 2025 10:44:29
We wish to thank the Editor and the Reviewers’ for their: again, we appreciate all your insightful comments, and we tried to be responsive to them. Modifications are highlighted in green.
Thank you for taking the time to help us to revise and improve our manuscript.
We look forward to hearing from you at your earliest convenience.
Sincerely,
Carmela Mento

Reviewer 2 Report
Comments and Suggestions for Authors
Dear authors,
Congratulation on the idea and method of your research. It is very original and innovative, in my opinion with not so much clinical applicability due to the need of experts in the method, the availability and the biomarkers now approved for the AD diagnosis and also the advanced imagery. I do have some comments:
- the authors and their affiliations are not mentioned in the manuscript and in the end their contributions
- row 36 - maybe rephrase , it does not make sense the ":"
- row 50 - you should mention a reference, and also a lot on these paragraphs emphasis on the costs of INPH rather than the clinical benefit
- your cohort is really small, but the methods and materials is very good explained and also the appendix are very useful and well explained for someone who is not an expert in ANNs
- it would have been useful to mention the condition of the patients (agitated, medicated already for anxiety or other psychiatric conditions) because it is useful considering that you are applying some test based on psychology
- the tables and figures are of great support
- I do not think that for rows 238-240 you actually need a reference for that very common phrase
- regarding the references: no 54,55,56 are self citations and need to be excluded; also the style of the reference writing is not the same in all citations; you have a lot of very very old citations and some of them refer to the same subject so please remove a few (mainly all of the refer to the Rorschach method and are from 1921 (no 37), 1936 (no 47), 1948 (no 48), 1945 (no 26), 1965 (no 1); i am sure you can find newer ones and if they are for historical purpose than please choose fewer
- also in references no 14 is about " Dementia in the elderly" from 1982, please find more accurate and up to date data; no 36 refers to "MMSE as a practical method for grading cognitive state of patients for the clinicians" from 1975!!; I am sure MMSE has been used a lot since then
Author Response
Review 2
Dear Editor,
Thank you for the opportunity to revise our above-referenced paper. The following letter gives our responses to the Editor’s and Reviewers’ comments, as requested.
We wish to thank the Reviewers for reading our manuscript and reviewing it, which has helped us improve it. We next detail our responses to each reviewer’s concerns and comments.
Open Review
(x) I would not like to sign my review report
( ) I would like to sign my review report
Quality of English Language
( ) The English could be improved to more clearly express the research.
(x) The English is fine and does not require any improvement.
|
Yes |
Can be improved |
Must be improved |
Not applicable |
|
|
Does the introduction provide sufficient background and include all relevant references? |
(x) |
( ) |
( ) |
( ) |
|
Is the research design appropriate? |
(x) |
( ) |
( ) |
( ) |
|
Are the methods adequately described? |
(x) |
( ) |
( ) |
( ) |
|
Are the results clearly presented? |
( ) |
(x) |
( ) |
( ) |
|
Are the conclusions supported by the results? |
(x) |
( ) |
( ) |
( ) |
|
Are all figures and tables clear and well-presented? |
(x) |
( ) |
( ) |
( ) |
Comments and Suggestions for Authors
Dear authors,
Congratulation on the idea and method of your research. It is very original and innovative, in my opinion with not so much clinical applicability due to the need of experts in the method, the availability and the biomarkers now approved for the AD diagnosis and also the advanced imagery. I do have some comments:
- the authors and their affiliations are not mentioned in the manuscript and in the end their contributions
- row 36 - maybe rephrase , it does not make sense the ":"
#R to Rev2#: We modified this part
- row 50 - you should mention a reference, and also a lot on these paragraphs emphasis on the costs of INPH rather than the clinical benefit
The text has been modified according to a recent conference at the Italian Ministry of Health, aimed at discussing this pathology and its underestimated costs.
- your cohort is really small, but the methods and materials is very good explained and also the appendix are very useful and well explained for someone who is not an expert in ANNs
- it would have been useful to mention the condition of the patients (agitated, medicated already for anxiety or other psychiatric conditions) because it is useful considering that you are applying some test based on psychology
R Rev 2: The patients did not present any mental disorders or behavioral changes except memory disturbances and underwent neurosurgical medical evaluation for hydrocephalus.
- the tables and figures are of great support
- I do not think that for rows 238-240 you actually need a reference for that very common phrase
R to Rev 2: the variables cited refer to elements classically described and assessed in the Rorschach personality test cited in the bibliography.
- regarding the references: no 54,55,56 are self citations and need to be excluded; also the style of the reference writing is not the same in all citations; you have a lot of very very old citations and some of them refer to the same subject so please remove a few (mainly all of the refer to the Rorschach method and are from 1921 (no 37), 1936 (no 47), 1948 (no 48), 1945 (no 26), 1965 (no 1); i am sure you can find newer ones and if they are for historical purpose than please choose fewer
- also in references no 14 is about " Dementia in the elderly" from 1982, please find more accurate and up to date data; no 36 refers to "MMSE as a practical method for grading cognitive state of patients for the clinicians" from 1975!!; I am sure MMSE has been used a lot since then
#R to Rev2#: Self citations have been removed and more recent references inserted.
Submission Date
09 June 2025
Date of this review
20 Jun 2025 07:16:41
We wish to thank the Editor and the Reviewers’ for their: again, we appreciate all your insightful comments, and we tried to be responsive to them. Modifications are highlighted in green.
Thank you for taking the time to help us to revise and improve our manuscript.
We look forward to hearing from you at your earliest convenience.
Sincerely,
Carmela Mento

Reviewer 3 Report
Comments and Suggestions for Authors
The introduction part is too long.
In such a low number of subjects, median values should be calculated rather than mean values.
Surgical outcome following diagnosis of INPH should also be given in means of those tests to validate pre-op outcomes correctly.
Author Response
Review 3
Dear Editor,
Thank you for the opportunity to revise our above-referenced paper. The following letter gives our responses to the Editor’s and Reviewers’ comments, as requested.
We wish to thank the Reviewers for reading our manuscript and reviewing it, which has helped us improve it. We next detail our responses to each reviewer’s concerns and comments.
Open Review
(x) I would not like to sign my review report
( ) I would like to sign my review report
Quality of English Language
( ) The English could be improved to more clearly express the research.
(x) The English is fine and does not require any improvement.
|
Yes |
Can be improved |
Must be improved |
Not applicable |
|
|
Does the introduction provide sufficient background and include all relevant references? |
(x) |
( ) |
( ) |
( ) |
|
Is the research design appropriate? |
(x) |
( ) |
( ) |
( ) |
|
Are the methods adequately described? |
(x) |
( ) |
( ) |
( ) |
|
Are the results clearly presented? |
(x) |
( ) |
( ) |
( ) |
|
Are the conclusions supported by the results? |
(x) |
( ) |
( ) |
( ) |
|
Are all figures and tables clear and well-presented? |
(x) |
( ) |
( ) |
( ) |
Comments and Suggestions for Authors
The introduction part is too long.
In such a low number of subjects, median values should be calculated rather than mean values.
Surgical outcome following diagnosis of INPH should also be given in means of those tests to validate pre-op outcomes correctly.
R Rev : the calculations have been described in the methodology of the text. This work deals with a pre-screening on patients who did not have medical outcomes.
Submission Date
09 June 2025
Date of this review
16 Jun 2025 13:15:12
- Please add the accuracy of names and affiliations.
#R to Rev 3#: We added names and affiliations.
- For abstract section, we strongly encourage authors to use the subheadings given. Background and Objectives, Materials and Methods, Results, and Conclusions.
#R to Rev 3#: The abstract has been divided into subheadings Background and Objectives, Materials and Methods, Results and Conclusions.
- : For research articles with several authors, a short paragraph specifying their individual contributions must be provided. The following statements should be used “Conceptualization, X.X. and Y.Y.; methodology, X.X.; software, X.X.; validation, X.X., Y.Y. and Z.Z.; formal analysis, X.X.; investigation, X.X.; resources, X.X.; data curation, X.X.; writing—original draft preparation, X.X.; writing—review and editing, X.X.; visualization, X.X.; supervision, X.X.; project administration, X.X.; funding acquisition, Y.Y. All authors have read and agreed to the published version of the manuscript.” Please turn to the CRediT taxonomy for the term explanation. Authorship must be limited to those who have contributed substantially to the work reported.
#R to Rev 3#: We done
- Discussion and Summary should be separated; please list the Discussion section separately.
#R to Rev 3#: We done
- Please provide us with the corresponding information.
Ethic Committee Name:
Approval Code:
Approval Date (Year. month. day.):
#R to Rev 3#: We added Ethic Committee Name
- Please submit the blank informed consent form used during the study.
#R to Rev 3#: We submitted the blank informed consent form
We wish to thank the Editor and the Reviewers’ for their: again, we appreciate all your insightful comments, and we tried to be responsive to them. Modifications are highlighted in green.
Thank you for taking the time to help us to revise and improve our manuscript.
We look forward to hearing from you at your earliest convenience.
Sincerely,
Carmela Mento

Round 2
Reviewer 1 Report
Comments and Suggestions for Authors
The manuscript is suitable for publication in its current form.